# Urinary Metals Concentrations and Biomarkers of Autoimmunity among Navajo and Nicaraguan Men

**DOI:** 10.3390/ijerph17155263

**Published:** 2020-07-22

**Authors:** Madeleine K. Scammell, Caryn Sennett, Rebecca L. Laws, Robert L. Rubin, Daniel R. Brooks, Juan José Amador, Damaris López-Pilarte, Oriana Ramirez-Rubio, David J. Friedman, Michael D. McClean, Johnnye Lewis, Esther Erdei

**Affiliations:** 1Department of Environmental Health, Boston University School of Public Health, Boston, MA 02118, USA; csennett@bu.edu (C.S.); rebecca.laws@gmail.com (R.L.L.); mmcclean@bu.edu (M.D.M.); 2Department of Molecular Genetics and Microbiology, University of New Mexico School of Medicine, Albuquerque, NM 87131, USA; RLRubin@salud.unm.edu; 3Department of Epidemiology, Boston University School of Public Health, Boston, MA 02118, USA; danbrook@bu.edu (D.R.B.); juanjoseamador3011@gmail.com (J.J.A.); dalp1342@gmail.com (D.L.-P.); oriana.ramirez@isglobal.org (O.R.-R.); 4Division of Nephrology, Department of Medicine, Beth Israel Deaconess Medical Center, Harvard Medical School, Boston, MA 02215, USA; dfriedma@bidmc.harvard.edu; 5College of Pharmacy, Community Environmental Health Program, University of New Mexico Health Sciences Center, Albuquerque, NM 87131, USA; jlewis@cybermesa.com (J.L.); EErdei@salud.unm.edu (E.E.)

**Keywords:** autoimmunity, antinuclear antibodies, specific autoantibodies, metals

## Abstract

Metals are suspected contributors of autoimmune disease among indigenous Americans. However, the association between metals exposure and biomarkers of autoimmunity is under-studied. In Nicaragua, environmental exposure to metals is also largely unexamined with regard to autoimmunity. We analyzed pooled and stratified exposure and outcome data from Navajo (*n* = 68) and Nicaraguan (*n* = 47) men of similar age and health status in order to characterize urinary concentrations of metals, compare concentrations with the US National Health and Nutrition Examination Survey (NHANES) male population, and examine the associations with biomarkers of autoimmunity. Urine samples were analyzed for metals via inductively coupled plasma mass spectrometry (ICP-MS) at the US Centers for Disease Control and Prevention. Serum samples were examined for antinuclear antibodies (ANA) at 1:160 and 1:40 dilutions, using an indirect immunofluorescence assay and for specific autoantibodies using enzyme-linked immunosorbent assay (ELISA). Logistic regression analyses evaluated associations of urinary metals with autoimmune biomarkers, adjusted for group (Navajo or Nicaraguan), age, and seafood consumption. The Nicaraguan men had higher urinary metal concentrations compared with both NHANES and the Navajo for most metals; however, tin was highest among the Navajo, and uranium was much higher in both populations compared with NHANES. Upper tertile associations with ANA positivity at the 1:160 dilution were observed for barium, cesium, lead, strontium and tungsten.

## 1. Introduction

In epidemiology, there is a long-standing interest in determining whether various diseases are concentrated in populations defined by racial, ethnic, demographic, and/or geographic characteristics. This is true of autoimmune diseases, a family of more than 80 human diseases, including type I diabetes, scleroderma, lupus, and multiple sclerosis. Autoimmune diseases affect between 14.7 and 23.5 million people in the US [1]. Epidemiological studies of autoimmune diseases have higher prevalence among indigenous populations compared with all North Americans, including higher rates of systemic lupus erythematosus (SLE) in the Algonkian First Nation community from Manitoba, Canada [2], and Crow, Arapahoe, and Sioux communities in the United States [3]; SLE and rheumatoid arthritis (RA) among Tlingit in Southeast Alaska [4]; and RA in US Yakima [5], Chippewa [6], and Pima [7] peoples.

Autoimmune diseases are suspected to be caused by a combination of environmental and genetic factors [8,9,10]. The use of diagnosed autoimmune disease as an outcome for epidemiological studies is often challenging, due to inconsistent, subjective, and/or changing diagnostic criteria, overlapping symptoms and co-morbidities, and the fact that there are no registries for these relatively rare diseases. An autoantibody measurement is one of the most useful serological tests for the detection and diagnosis of many systemic and organ-specific autoimmune diseases, as well as for monitoring subclinical, autoimmune-driven perturbations that do not necessarily result in disease [11]. In addition, autoantibody biomarkers can precede development of overt clinical disease by many years [12,13,14], thereby providing information potentially useful for early detection and therapeutic intervention of autoimmune disease [15,16]. 

Exposures to metals and autoimmune outcomes are not well understood [17]. A high concentration of some metals may overload the immune system, while conversely, deficiencies in metals with nutritional value may also trigger autoimmune response [17]. Metals exposures from occupational and natural sources are hypothesized to play a role in autoimmune conditions. Rodent studies suggest an association between metals exposure and autoantibody production [18,19], and some epidemiological studies have identified arsenic and mercury (separately, not combined) as predictors of serum autoantibodies [20] and subsequent autoimmune disease development [21,22,23]. Molecular mechanisms are likely metal-induced generation of reactive oxygen species [24], followed by oxidative damage to tissues, organs and the genetic material [25]. Metals may also directly enhance T lymphocyte activation [26] causing hyperreactivity to self-antigens. However, there is insufficient evidence in humans to support the role of metals in the causation of autoimmune disease [27]. An expert panel convened by the National Institutes of Health describes an understanding of metals and autoimmunity as a “major research gap” [28], recommending more research in this area, with a particular emphasis on biomarkers of exposure (e.g., urinary metals) and effect (e.g., specific autoantibodies). 

Metals exposure among indigenous populations has been previously documented [29]. From 1944 to 1986, mining operations removed more than 30 million tons of uranium ore in the Navajo Nation that left the lasting legacy of abandoned uranium mines, milling sites, groundwater containing uranium above safe levels causing environmental and public health concerns to Navajo communities [30]. Exposure to mixed-metal wastes resulting from uranium and other hard-rock mining activities on tribal lands is ongoing [31,32]. While, the exposure to mercury from fish consumption and residential proximity to an arsenic-contaminated site have been associated with increased prevalence of antinuclear antibodies (ANA). Specific autoantibodies [33] in a Sioux community, and elevated autoantibodies, are associated with living near uranium mines in Navajo Nation residents [34]. There are no studies so far that have assessed the relationship of in vivo biomarkers of autoimmunity with a comprehensive panel of metals yet. 

In multiple Central American countries, high concentrations of arsenic (>50 µg/L) have been measured in common drinking water sources [35] and arsenic, cadmium, uranium, strontium, barium, manganese, and lead have been detected in volcanic emissions and in soils surrounding active volcanoes in the region [36,37,38,39,40]. Data on autoimmune disease in Central America are lacking [10]. Both human exposure to metals and serum ANA have been measured infrequently in epidemiological studies in Central America [41,42]. One case series study of Nicaraguan patients (*n* = 19) with Mesoamerican Nephropathy (a chronic kidney disease of unknown etiology) reported a single patient with serum ANA positivity but no specific autoantibodies for double-stranded DNA, centromere antibodies or extractable nuclear antigens [41]. The same study reported detectable urinary concentrations of lead, cadmium, uranium and mercury, but did not assess the association between metals exposure and autoimmune biomarkers [41].

In addition to shared concerns regarding metals exposures among an indigenous North American and a Central American community, increasing prevalence of chronic diseases relative to US white populations have been reported among Navajo Nation and in Mesoamerica (including Nicaragua) [43,44,45,46]. These epidemiological similarities, as well as the availability of both urinary metals and ANA data in both populations, prompted the current collaborative, hypothesis-generating study to investigate the relationships between environmental metal contaminants and serological biomarkers of autoimmune disease in groups from both regions.

Using data from two geographically distinct groups, men from the Navajo Birth Cohort Study (NBCS) and men from an occupational study of sugarcane workers in Nicaragua, the present study had the following objectives: (1) Characterize exposure to metals among healthy Nicaraguan and Navajo men of working age; (2) evaluate biomarkers of autoimmunity (ANA and specific autoantibodies) in each group; and (3) examine the relationship between metals exposure and biomarkers of autoimmunity in the pooled population.

## 2. Materials and Methods 

### 2.1. Study Population

The study population included 47 male sugarcane workers from northwestern Nicaragua and 68 fathers from the Navajo Nation. All study participants were at least 18 years of age. Previously, we described the enrollment of 284 sugarcane workers from one company in the department of Chinandega, Nicaragua, in 2010 [47]. In March 2015, we re-sampled 50 of these same workers randomly selected from among those who were still employed by the company in 2015 and worked in one of four job tasks: (1) Cane cutting, (2) irrigation, (3) seeding/seed-cutting or (4) agrichemical application. We collected blood and urine samples post-shift. Three workers refused the blood draw, resulting in complete data for 47 participants. All workers completed questionnaires reporting no disease diagnoses, chronic or acute health conditions. 

The NBCS is a congressionally mandated collaborative research study originally supported by the Agency for Toxic Substances and Disease Registry (CDC/ATSDR) with research led by the University of New Mexico Community Environmental Health Program in partnership with the Southwest Research and Information Center, the CDC Division of Laboratory Sciences, the Navajo Nation Department of Health and other Navajo agencies, the Navajo Area Indian Health Service (IHS), and the previously acknowledged six IHS and PL-638 healthcare facilities on Navajo Nation. The purpose of the study is to investigate birth outcomes and child development through age one year in relation to non-occupational exposures to uranium wastes from past mining and milling operations on the Navajo Nation. This study encourages father participation. Nearly 200 NBCS fathers were recruited between February 2013 and December 2015 in six geographic areas of Navajo Nation representing mined and unmined regions. At enrollment, fathers provided blood and urine samples for clinical and research analyses, including investigation of the association between metals exposure, immune function, and autoimmunity. A subset of sixty-eight of these fathers were selected for the current analysis by matching age and health status to the previously recruited Nicaraguan workers, all of whom were men. Due to these selection criteria, the Navajo fathers in this study are not representative of the NBCS fathers. 

The Institutional Review Boards at the Boston University Medical Campus and the Nicaraguan Ministry of Health approved the study protocols for the studies in Nicaragua. NBCS received approvals from the University of New Mexico Health Sciences Center Human Research Protections Office (11-310) and the Navajo Nation Human Research Review Board (NNR 11.323), which continues oversight of the project. All participants provided informed consent prior to participation in research activities.

### 2.2. Laboratory Analysis of Urinary Metals

All urine samples from both studies were analyzed for metals at the US Centers for Disease Control and Prevention Division of Laboratory Science in Atlanta, Georgia, USA. Metals (antimony, total arsenic, barium, cadmium, cesium, cobalt, lead, manganese, molybdenum, strontium, thallium, tin, tungsten, and uranium) were analyzed using an inductively coupled plasma dynamic reaction cell mass spectrometer (ICP-DRC-MS) (PerkinElmer NexION 300D, Waltham, USA) [48]. Speciated arsenic (arsenobetaine, arsenocholine, arsenite (As III), arsenate (As V), monomethylarsonic acid (MMA), and dimethylarsinic acid (DMA)) was also determined using inductively coupled mass spectrometry (ICP-MS) coupled with high performance liquid chromatography [49]. Limits of detection are published in Appendix A. Urine creatinine was analyzed via enzymatic method using the Roche/Hitachi Modular P Chemistry Analyzer.

### 2.3. Laboratory Analysis of Biomarkers of Autoimmunity

The presence of antinuclear antibody (ANA) was determined by indirect immunofluorescence (IIF) at the University of New Mexico Health Sciences Center using Hep2 cells as substrate and fluorescein-labeled anti-IgG (H + L) as the detecting reagent (INOVA Diagnostics, San Diego, CA, USA). Sera were diluted 1:40 and 1:160 in phosphate-buffered saline. A dilution of 1:80–1:160 is standard for clinical studies [50,51,52,53], while the 1:40 dilution is intended for exploratory, hypothesis generating [54] studies. Slides were viewed independently by two observers (E.E. & R.L.R.) using an Olympus fluorescence microscope at 600-fold magnification (Olympus, Lake Success, NY, USA). Staining patterns (nuclear, nucleolar, cytoplasmic) were graded on a scale of 0–4+ intensity, based on positive and negative control standard sera that were run with each assay.

Specific autoantibodies were also detected by enzyme-linked immunosorbent assay (ELISA) using an in-house procedure, as previously described [55,56], and used by members of our team in Navajo Nation studies [34]. Antibody binding to chromatin, histones, denatured DNA (dDNA or single-stranded DNA), and native DNA (nDNA or double-stranded DNA) was quantified in duplicate serum samples in a 1:200 dilution after incubation in antigen-coated wells for 2 h at room temperature [55,56]. Bound antibodies were detected with peroxidase-conjugated anti-human IgG (Southern Biotech Associates, Birmingham, AL, USA) followed by color development using a peroxidase secondary substrate and reported as optical density (O.D.) units. Positive and negative controls were included in each assay. Two standard deviations above the mean of the negative controls was considered elevated reactivity.

### 2.4. Statistical Analysis

We created a dichotomous outcome variable for each serum dilution (1:160 and 1:40) to categorize ANA positivity. Positivity was defined as a ≥2+ intensity fluorescence score, which is an appropriate cut-off for a healthy population without known autoimmune disease [57]. To account for differences in urine concentration among individuals, we ran two distinct models. In the first, we normalized metal concentrations to urinary creatinine concentration (g/L), as is conventional practice. In the second model, we statistically adjusted for urinary creatinine as a variable in our regression models, as suggested by Barr et al. 2005 [58], to account for population differences in urine concentration and time of day of urine collection. Metal values below the limit of detection (LOD) were substituted with LOD/√2. Median concentrations of each metal were compared across groups (Navajo and Nicaraguan men) and to the United States adult male population (20 years and older) using data from the 2011–2012 National Health and Nutrition Examination Survey (NHANES), conducted by the US Centers for Disease Control and Prevention [59]. 

ANA and ELISA autoantibody positivity were first characterized for each group. We then assessed the association between each metal and ANA positivity using multivariable logistic regression. We pooled data on the 47 Nicaraguan and 68 Navajo men, creating more power in our sample (*N* = 115), and examined the spread of metals concentrations in the pooled data, calculating exposure tertiles for each metal (normalized to urine creatinine). Metals tertile, the categorical exposure variable, was used to predict an aggregate outcome variable indicative of any ANA response at one or more of the three cellular sites (positivity at the cytoplasmic, nuclear or nucleolar site) at the 1:40, and 1:160 dilutions, respectively. 

Additional predictors included in all models were age, group (Navajo or Nicaragua) and seafood consumption within the past three days (yes/no). Recent seafood consumption data were lacking for the Navajo participants, however, over 60% of the NBCS mothers reported never having consumed fish, ever. Given cultural beliefs and geographic location in the high desert of Southwestern US, which both preclude consumption of seafood, Navajo participants were coded as unexposed. Data on current cigarette smoking were also lacking among Navajo participants. However, among the Nicaraguan participants current smoking behavior was predictive of increased metals exposure, but was not associated with ANA positivity. Consequently, we ruled out smoking behavior as a potential confounder in our pooled analyses. Given the hypothesis-generating nature of this study, the results were evaluated, based on the strength of association, rather than solely on statistical significance. As such, multiple comparison correction was overly conservative and the data were incompatible with the assumptions of penalized regression models. Data were analyzed using Statistical Analysis Software version 9.4 (SAS Institute Inc. Cary, NC, USA).

## 3. Results

### 3.1. Population Characteristics

All participants were male with an average age of 32 years (range: 23–51 years) (Table 1). The Nicaraguan group was composed of workers employed in active jobs that required them to pass a health screening. Questionnaires also confirmed health status. The selected Navajo group reported having no diagnosed diseases including cardiovascular disease, high blood pressure, diabetes or kidney disease.

### 3.2. Urinary Metals

All urine samples had detectable levels of barium, cesium, cobalt, lead, molybdenum, and strontium (Table 1). We calculated the arithmetic median concentration and range of each metal in both groups and the pooled cohort (Table 1). We also calculated the geometric means and 95% confidence limits for each metal in in both groups, comparing these results with US males in NHANES (Figure 1 and Figure 2). Concentrations of most metals were higher among the Nicaraguan men compared to concentrations among Navajo men (Table 1), and compared to the US adult males (Figure 1 and Figure 2). Median concentrations of arsenic, barium and strontium were approximately twice as high among the Nicaraguan men as compared to US adult male geometric mean concentrations in NHANES and the Navajo men (Figure 1, Figure 2 and Table 1). The concentrations of all arsenic species, cesium, cobalt, lead, manganese, molybdenum and uranium were also higher among the Nicaraguans, as compared to NHANES and the Navajo. Metals’ concentrations that were lower among Nicaraguan men, compared with NHANES and Navajo men were antimony, cadmium, thallium, tin and tungsten. Antimony, cadmium, tin and tungsten were all highest among the Navajo compared with both NHANES and the Nicaraguans, with tin notably higher and detected in 99% of samples among the Navajo as compared to 48% among the Nicaraguans. Uranium concentrations were somewhat similar between the Nicaraguan (0.02 μg/g creatinine) and Navajo (0.01 μg/g creatinine) groups, although 2–3-fold higher compared to NHANES (0.004 μg/g creatinine) (Figure 1 and Figure 2).

### 3.3. Autoimmune Biomarkers

Any ANA positivity based on a staining intensity of ≥2+ at the 1:160 serum dilution was similar between the two groups (7.4% among the Navajo; 8.5% among the Nicaraguans) (Table 2). ANA positivity at the nuclear site was also similar between groups at both dilutions. However, cellular site specific staining patterns differed between the two groups; this was most apparent at the 1:40 dilution in which the prevalence of ANA positivity was naturally higher than at the 1:160 dilution. Any ANA positivity was higher among the Nicaraguans (80.9%), compared to the Navajo (55.9% positivity), although nucleolar staining was detected in 12% of the Navajo samples at 1:40 dilution, but in none of the Nicaraguan samples at either serum dilution. In contrast, the Nicaraguans had more anti-cytoplasmic (74.5%) antibody reactivity, as compared to the Navajo (47.1%) at the 1:40 dilution. Overall, in the pooled cohort (*n* = 115), 66.1% were ANA positive at any cellular site at the 1:40 serum dilution compared to 7.8% positive at any cellular site at the 1:160 serum dilution (Table 2).

Specific autoantibodies measured by ELISA are shown in Table 2 and Figure 3. Prevalence of reactivity to the tested antigens was low. In the Nicaraguan group, three participants had slightly elevated reactivity to dDNA, one to histones and one to chromatin; in the Navajo group, low-level reactivity to dDNA was detectable in seven participant samples and to the other antigens (chromatin, histones and nDNA) in two participants.

### 3.4. Associations between Metals and ANA Positivity in Pooled Analyses (Table 3)

It was not possible to run the models, including urine creatinine as a co-variate for several of the arsenic metabolites, due to low percent detection. Instead we focus discussion on the results of the model evaluating creatinine-normalized concentrations, but provide the results of both normalizing creatinine, and including it as a variable in Appendix A, Associations between urinary metal concentrations and ANA positivity differed at the two dilutions (1:160 and 1:40). At the 1:160 dilution, odds of ANA positivity at any cellular site were highest and significant among participants with third tertile exposure as compared to participants with first tertile exposure for cesium (OR = 2.98, 95% CI: 1.07, 8.25), lead (OR = 3.31, 95% CI: 1.09, 9.97), strontium (OR = 4.71, 95% CI: 1.34, 16.61) and tungsten (OR = 4.00, 95% CI: 1.11, 14.44) (Table 3). Third tertile barium exposure also had a high association, although not significant (OR = 2.33, 95% CI: 0.83, 6.55). Lower (OR’s ranging from 1.01 to 1.78), albeit positive associations between any ANA positivity and third tertile concentrations were also observed for total arsenic, arsenobetaine, arsenocholine, As (III), As (V), antimony, cadmium, cobalt, thallium, tin and uranium. Although the odds of any ANA positivity were higher among participants with second tertile urinary concentrations of DMA (OR = 1.46, 95% CI: 0.56, 3.82), there was no association between the third tertile exposure to DMA and ANA positivity (OR = 0.45, 95% CI: 0.10, 1.97). No association was seen between urinary concentrations of MMA and ANA positivity at the 1:160 dilution.

The results at the 1:40 dilution were somewhat disparate in that the only comparably high odds ratios for ANA associated with metals in third tertile of exposure were for urinary DMA (OR = 2.17, 95% CI: 0.85, 5.47), which was an opposite trend as at the 1:160 dilution. Although, neither was significant. Associations at 1:40 dilution between any ANA positivity and third tertile urinary concentrations of total arsenic, arsenobetaine, cobalt, strontium and thallium were also positive, but lower—similar to the results seen at the 1:160 dilution (OR’s ranged from 1.08 to 1.72). However, antimony in the second tertile exposure was associated with any ANA positivity at the 1:40 dilution (OR = 2.25, 95% CI: 1.16, 4.35), third tertile exposure was not (Table 3). This was also true at the 1:160 dilution. At the 1:40 dilution, there was no association between urinary tin or uranium and ANA positivity. Finally, the odds ratios were undefined given low counts in each tertile for urinary molybdenum and manganese at the 1:160 dilution and are not included in Table 3.

## 4. Discussion

This is the first biomonitoring study to examine the prevalence and associations between urinary concentrations of metals and autoimmune biomarkers in Nicaraguan or Navajo men. With respect to metals, urinary concentrations of most metals assessed in this study were highest among the Nicaraguan group. As Nicaragua is a volcanic region that is high in strontium [38], lava and bedrock are probable environmental sources and may explain the higher urinary concentrations of strontium in the Nicaraguan group (median 222.28 μg/g creatinine), as compared to the Navajo group (median 130.16 μg/g creatinine) and NHANES (median 91.8 μg/g creatinine). Volcanic emissions are also known to contain barium, manganese [57], and arsenic [35]. The high urinary concentrations of total arsenic among the Nicaraguans (median 13.5 μg/g creatinine, as compared to NHANES median of 6.1 μg/g creatinine) may also reflect the presence of arsenic in drinking water sources in the region [35]. Dietary habits, such as seafood consumption (as evidenced by the high concentrations of the organic arsenic species arsenobetaine and DMA relative to inorganic arsenic species) [59] or exposure to arsenical pesticides historically used in the region. The low arsenobetaine concentrations among the Navajo are consistent with a population that does not consume seafood [60,61]. It is notable, however, that the inorganic arsenic species are high among the Navajo compared with NHANES.

Navajo participants were expected and found to have relatively high urine uranium, due to proximity to abandoned uranium mine and milling sites. Although, they were not selected with any consideration of exposure history, but rather with a focus on health status. It is possible that the even higher uranium concentrations in Nicaraguan participants may be due to the volcanic activity in the area, but uncertain at best.

While antimony, cadmium, and tungsten concentrations were slightly higher among the Navajo as compared with NHANES and the Nicaraguan men, urinary concentrations of most metals in the Navajo group were lower or similar to NHANES, with the noted exceptions of tin and uranium. A subsequent analysis of urine metals compared Navajo men selected for this study to Navajo men in the larger Navajo Birth Cohort Study (NBCS). The results indicated that there was no difference in urinary concentrations between the subgroups with the exception of tin. Urine tin concentrations from NBCS men selected for analysis were marginally greater (Mann-Whitney U test, *p*-value 0.045) than concentrations among NBCS men who were not selected for this study. Compared to the NHANES 2011–2012 report of median tin levels among adult men, urinary tin concentrations were high in the Navajo group and 15-fold higher than in the Nicaraguan group. While, tin exposure can occur via contact with dust and soil containing tin [62], the primary exposure route is thought to be via the consumption of food and beverages from tin-containing cans and food packaging [63].

The prevalence of any ANA positivity in the current sample is similar to that observed in other studies of healthy individuals suggesting the findings in this study are generalizable beyond our somewhat unique exposure scenarios. The prevalence of ANA positivity (8%) in the present male population with an average age of 32 years is also similar to the ~8% prevalence of ANA positivity among the male US NHANES population age 30–39 years (1999–2004) [64]. Satoh et al. [64] used a cut point of a 3+ intensity score and a dilution of 1:80, while we used a lower intensity ANA score (2+) but higher serum dilution (1:160) that may be more clinically relevant. Dinse et al. [16] estimated 11% prevalence among adult males during the same time period as Satoh et al. [64] at 1:80 dilution and 3+ ANA intensity.

The dilution (1:160 v. 1:40) had a substantial effect on our results. In the present study, two-thirds more of the pooled population had positive antibody binding to any cellular site at the 1:40 dilution compared with the 1:160 dilution. Qualitatively similar observations were made by Mariz et al. [65] in healthy Brazilians where the percentage of ANA positivity increased from 7.6% at 1:160 dilution to 46% at a lower (1:80) dilution. Similarly, Marin et al. [66] found 35.4% of healthy individuals had nuclear staining at 1:40 dilution, and at the 1:160 dilution the same population had only 3.2% staining.

The 1:40 dilution is more sensitive and useful for research examining possible associations with environmental exposures, but has a higher possibility of false positives. The basis for serum reactivity at lower dilution is due, in part, to autoantibodies of the lens epithelium-derived growth factor (LEDGF) nuclear antigen, which has no clinical significance [65,67], and to non-specific binding at high concentration of polyclonal immunoglobulin in this assay format. Nevertheless, the finding that third tertile exposures to DMA, arsenic, arsenobetaine, cobalt, strontium, and thallium were associated, albeit weakly, with ANA at the 1:40 dilution suggests that, long-term exposure to relatively low levels of these metals, may promote production of autoimmune biomarkers.

Specific autoantibodies tested by ELISA in this study (histones, dDNA, chromatin and nDNA) were observed infrequently and only at low levels among Nicaraguan and Navajo men. While, antibodies to histones and dDNA are sensitive indicators of xenobiotic-induced antibodies, especially by ingested medications [68]. Similar immune perturbations did not occur to environmental metals at the exposure duration and levels encountered in the current study. Antibodies to chromatin and nDNA, considered indicators of idiopathic autoimmune disease, were detected at low levels only in two Navajo men. The lack of a strong response to these antigens is consistent with the health status of the groups studied.

Based on the strength of association, the metals that were seemingly associated with ANA positivity at the 1:160 dilution in the present study are barium, cesium, lead, strontium and tungsten. Although, other metals were also positively associated with ANA, the associations were weak or close to null. Lead was the only metal examined in our study with toxicological literature supporting the association [19,69]. However, a recent epidemiological analysis by Dinse et al. [16] found no association between blood lead in NHANES and ANA among men and women, a result which is inconsistent with those of the present study. No literature was found examining the associations of barium, cesium, strontium or tungsten with ANA or any autoimmune outcomes in humans or in animals. Although, there is literature suggesting an association between exposure to arsenic [23,70] and cadmium [18,71], and biomarkers of autoimmunity, the associations observed in the present study were positive but weak. Future investigations are needed to ascertain the relationship between arsenic, cadmium and the autoimmune biomarker production. 

Overall, the results of our analyses suggest a possible association between exposures to certain metals (barium, lead, strontium, cesium and tungsten) and the production of antinuclear antibodies, when assessed for positivity at the 1:160 dilution. Given the possibility of interactions between metals, additional research assessing the impact of metal mixtures on autoimmune outcomes is warranted.

This study has several limitations. Although, the prevalence of ANA positivity at the 1:160 dilution is consistent with previously published population-based, healthy or undiagnosed patients’ studies [57,64], the small sample size, and relatively low prevalence of positive outcomes (ANA and specific autoantibody reactivity), requires cautionary interpretation of statistical significance. We did not include body weight as a variable in our models, which could influence urinary metals concentrations. The small sample size also limited our ability to assess interactive effects between metals or the association between metals and specific autoantibodies. The present study was also limited to quantifying a small selection of autoantibodies; future studies should consider inclusion of a broader range of autoantibodies specific to different target tissues such as thyroid or pancreatic antigens. The participants in this study were exclusively male, and neither the healthy Nicaraguan workers, nor the Navajo men selected for health status reflect their populations as a whole. Given that women have higher autoimmunity biomarker production, and are most affected by autoimmune disease [64], future studies examining environmental exposures and autoimmune biomarkers should include female participants [1]. Finally, only single urine measures at one point in time were collected for all participants. No diurnal or seasonal variation in urine metal concentrations were examined, and we were not able to examine mercury exposure among the metals.

## 5. Conclusions

The low prevalence of ANA positivity in this pooled cohort of relatively young, healthy men using a conservative assay cut-off criterion is consistent with rates published in the literature. Similarly, reactivity to specific autoantibodies (histone, dDNA, nDNA and chromatin) was low. We found urinary concentrations of arsenic, barium and strontium were twice as high among Nicaraguan men, compared with the US population as measured by NHANES, and manganese 50% higher in the Nicaraguan group as compared to the Navajo group. Other adverse outcomes that we did not study may be associated with these high urine concentrations, and follow-up on exposure sources may be warranted. Notable associations between ANA positivity at the 1:160 dilution and upper tertile concentrations of several metals including barium, cesium, lead, strontium, and tungsten suggest that more research on metals and autoimmune biomarkers may also be warranted.

## Figures and Tables

**Figure 1 ijerph-17-05263-f001:**
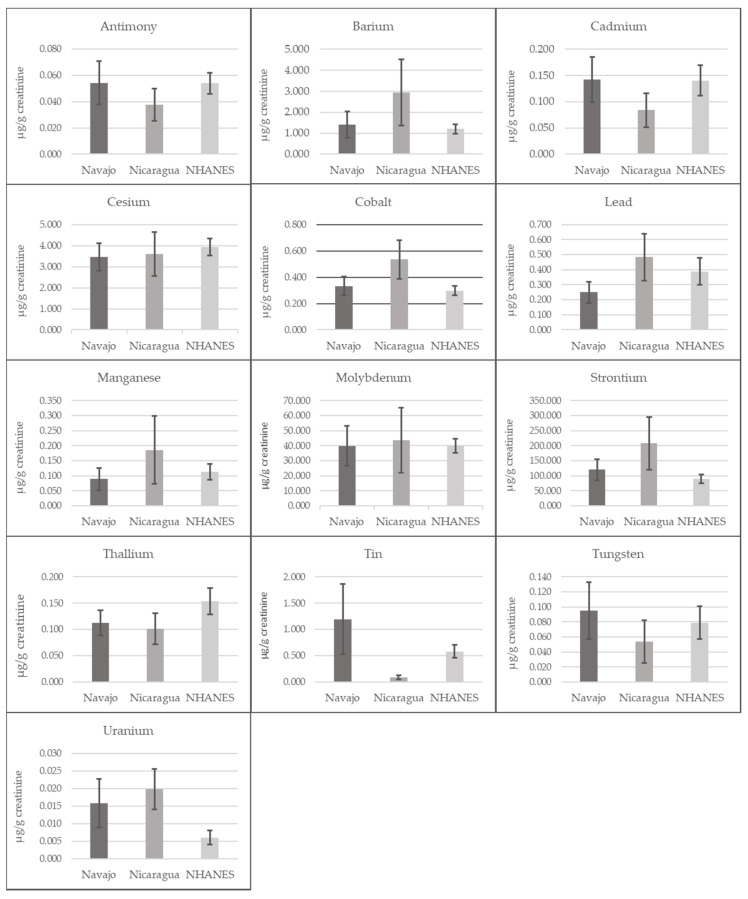
Distribution of geometric mean metal concentrations (μg/g creatinine) and 95% confidence limits in comparison with geometric mean concentrations of the United States adult male population 2011–2012. Note: US metals reported in the 2011–2012 National Health and Nutrition Examination Survey (NHANES) Fourth Report [59].

**Figure 2 ijerph-17-05263-f002:**
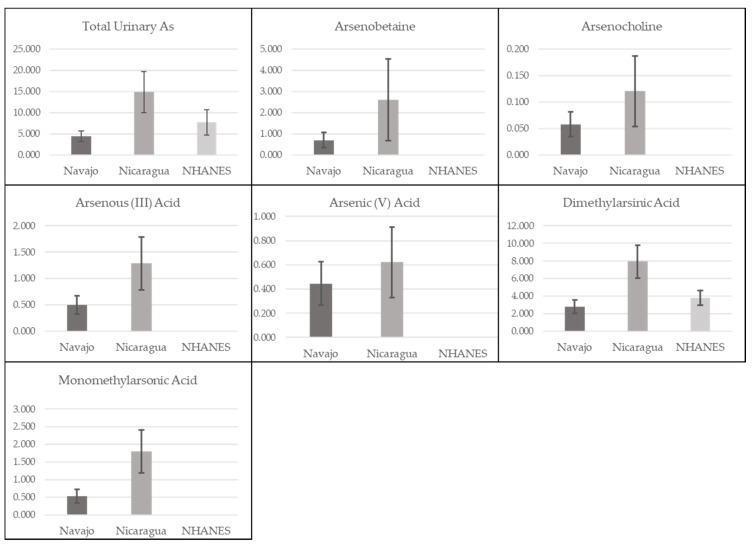
Distribution of geometric mean total and speciated arsenic concentrations (μg/g creatinine) and 95% confidence limits in comparison with geometric mean concentrations of the United States adult male population 2011–2012 [59]. Note: Arsenic is reported in μg/L arsenic.

**Figure 3 ijerph-17-05263-f003:**
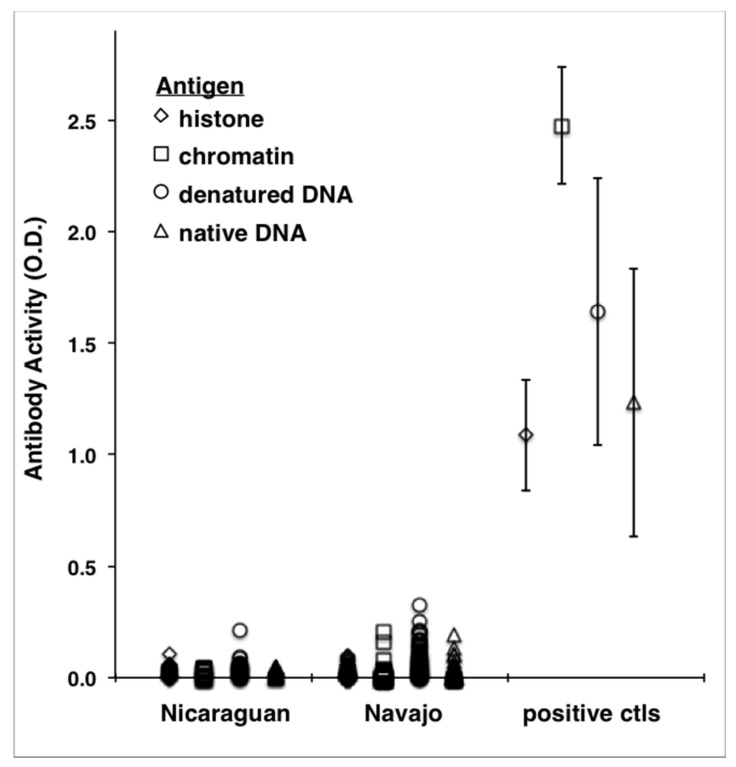
ELISA autoantibody activity in optical density units (O.D.) among Nicaraguan and Navajo men, and comparison with positive controls. IgG binding to the indicated antigens was measured in five separate assays. The mean ± SD of positive control samples included in each assay is shown.

**Table 1 ijerph-17-05263-t001:** Demographics and urinary metals detection and concentrations (μg/g creatinine) among the study population.

Variable	Pooled (*n* = 115)			Navajo (*n* = 68)			Nicaraguan (*n* = 47)		
Demographics									
Male Sex, *n* (%)	115 (100%)			68 (100%)			47 (100%)		
Median Age, years (range)	31.7 (23–51)			31 (26–47)			32.2 (23–51)		
Seafood Consumed past 3 days, *n* (%) ^a^	-			-			12 (23.40%)		
Urinary Metals	% Detect	Median (μg/g)	Range (μg/g)	% Detect	Median (μg/g)	Range (μg/g)	% Detect	Median (μg/g)	Range (μg/g)
Total Arsenic	99.5%	6.32	(2.04–66.55)	99%	4.63	(2.04–25.51)	100%	13.50	(4.06–66.55)
Arsenobetaine	34%	0.88	(0.18–46.59)	8%	0.60	(0.18–17.92)	60%	2.04	(0.27–46.59)
Arsenocholine	24%	0.07	(0.02–1.03)	3%	0.06	(0.02–0.88)	45%	0.11	(0.02–1.03)
As (III)	95.5%	0.77	(0.09–5.56)	97%	0.54	(0.09–1.74)	94%	1.17	(0.27–5.56)
As (V)	24.5%	0.47	(0.11–6.29)	6%	0.40	(0.18–6.29)	43%	0.54	(0.11–4.44)
Dimethylarsonic Acid (DMA)	89.5%	4.18	(0.78–19.76)	90%	2.86	(0.78–15.21)	89%	7.57	(2.54–19.76)
Monomethylarsonic Acid (MMA)	93.5%	0.87	(0.16–9.31)	91%	0.55	(0.16–2.41)	96%	1.77	(0.54–9.31)
Antimony	87%	0.05	(0.01–0.36)	97%	0.05	(0.02–0.36)	77%	0.03	(0.01–0.20)
Barium	100%	2.03	(0.21–15.49)	100%	1.52	(0.21–12.47)	100%	3.06	(0.43–15.49)
Cadmium	88%	0.12	(0.01–0.62)	99%	0.14	(0.02–0.62)	77%	0.09	(0.01–0.26)
Cesium	100%	3.59	(0.55–8.75)	100%	3.35	(1.64–8.75)	100%	3.85	(0.55–8.6)
Cobalt	100%	0.42	(0.13–1.55)	100%	0.34	(0.13–1.55)	100%	0.55	(0.13–1.45)
Lead	100%	0.31	(0.09–2.36)	100%	0.24	(0.09–0.70)	100%	0.47	(0.14–2.36)
Manganese	58%	0.12	(0.03–5.81)	35%	0.1	(0.03–1.04)	81%	0.18	(0.03–5.81)
Molybdenum	100%	44.24	(1.87–213.89)	100%	39.89	(13.81–146.8)	100%	54.58	(1.87–213.89)
Strontium	100%	151.29	(23.85–726.83)	100%	130.16	(23.85–316.58)	100%	222.28	(33.84–726.83)
Thallium	98.5%	0.11	(0.02–0.28)	99%	0.12	(0.04–0.28)	98%	0.10	(0.02–0.27)
Tin	73.5%	0.38	(0.02–6.54)	99%	1.22	(0.08–6.54)	48%	0.08	(0.02–0.51)
Tungsten	91.5%	0.07	(0.01–1.80)	96%	0.10	(0.02–0.51)	87%	0.05	(0.01–1.80)
Uranium	98%	0.02	(0.004–0.10)	100%	0.01	(0.004–0.10)	96%	0.02	(0.01–0.08)

^a^. No seafood consumption data for NBCS group.

**Table 2 ijerph-17-05263-t002:** Biomarkers of autoimmunity among the study population.

Biomarkers	Pooled (*n* = 115)		Navajo (*n* = 68)		Nicaraguan (*n* = 47)	
ANA Positivity (≥2), 1:160 dilution ^a^	*n*	(%)	*n*	(%)	*n*	(%)
Any Site	9	(7.8%)	5	(7.4%)	4	(8.5%)
Nuclear Site	5	(4.4%)	3	(4.4%)	2	(4.3%)
Cytoplasmic Site	3	(2.6%)	1	(1.5%)	2	(4.3%)
Nucleolar Site	1	(0.9%)	1	(1.5%)	0	(0%)
ANA Positivity (≥2), 1:40 dilution ^a^	*n*	(%)	*n*	(%)	*n*	(%)
Any Site	76	(66.1%)	38	(55.9%)	38	(80.9%)
Nuclear Site	33	(28.7%)	19	(27.9%)	14	(29.8%)
Cytoplasmic Site	67	(58.3%)	32	(47.1%)	35	(74.5%)
Nucleolar Site	8	(7.0%)	8	(11.8%)	0	(0%)
Specific AutoAntibody (SpAuAb) Elevation	*n*	(%)	*n*	(%)	*n*	(%)
Any SpAuAb	13	(11.3%)	9	(13.2%)	4	(8.5%)
Histone	3	(2.6%)	2	(2.9%)	1	(2.1%)
Chromatin	3	(2.6%)	2	(2.9%)	1	(2.1%)
dDNA	10	(8.7%)	7	(10.3%)	3	(6.4%)
nDNA	2	(1.7%)	2	(2.9%)	0	(0%)

Note: Results are presented as *n* (%). ^a^ Antibody reactivity (“ANA”) at various intracellular sites was determined on serum samples via indirect immunofluorescence at two dilutions, results of which are listed separately.

**Table 3 ijerph-17-05263-t003:** Multivariable logistic regression results: Odds of association between urinary metals concentrations (μg/g creatinine) by tertile with first tertile as reference (REF) and any ANA positivity (≥2+) at cellular sites.

Metals Tertiles	Any ANA Positivity (1:160 dilution)	Any ANA Positivity (1:40 dilution)
Metals	OR (95%CI)	OR (95% CI)
Total Arsenic—Tertile 1 (4.67μg/g)	REF	REF
Total Arsenic—Tertile 2 (11.14 μg/g)	0.64 (0.21, 1.97)	1.01 (0.55, 1.86)
Total Arsenic—Tertile 3 (66.55 μg/g)	1.69 (0.41, 6.82)	1.72 (0.68, 4.31)
Arsenobetaine—Tertile 1 (0.60 μg/g)	REF	REF
Arsenobetaine—Tertile 2 (1.52 μg/g)	0.76 (0.25, 2.32)	0.78 (0.43, 1.42)
Arsenobetaine—Tertile 3 (46.59 μg/g)	1.02 (0.30, 3.42)	1.55(0.72, 3.35)
Arsenocholine—Tertile 1 (0.05 μg/g)	REF	REF
Arsenocholine—Tertile 2 (0.10 μg/g)	0.73 (0.24, 2.23)	1.13 (0.63, 2.03)
Arsenocholine—Tertile 3 (1.03 μg/g)	1.11 (0.38, 3.25)	0.95 (0.49, 1.82)
As (III)—Tertile 1 (0.51 μg/g)	REF	REF
As (III)—Tertile 2 (1.08 μg/g)	1.10 (0.32, 3.71)	1.39 (0.76, 2.53)
As (III)—Tertile 3 (5.56 μg/g)	1.01 (0.41, 2.48)	0.48 (0.23, 1.03)
As (V)—Tertile 1 (0.36 μg/g)	REF	REF
As (V)—Tertile 2 (0.60 μg/g)	1.05 (0.38, 2.92)	1.40 (0.77, 2.56)
As (V)—Tertile 3 (6.29 μg/g)	1.17 (0.42, 3.29)	0.73 (0.39, 1.35)
DMA—Tertile 1 (2.95 μg/g)	REF	REF
DMA—Tertile 2 (6.37 μg/g)	1.46 (0.56, 3.82)	0.78 (0.43, 1.43)
DMA—Tertile 3 (19.76 μg/g)	0.45 (0.10, 1.97)	2.17 (0.85, 5.47)
MMA—Tertile 1 (0.57 μg/g)	REF	REF
MMA—Tertile 2 (1.32 μg/g)	0.98 (0.37, 2.61)	1.20 (0.65, 2.23)
MMA—Tertile 3 (9.31 μg/g)	0.93(0.22, 3.86)	1.08 (0.42, 2.75)
Antimony—Tertile 1 (0.03 μg/g)	REF	REF
Antimony—Tertile 2 (0.06 μg/g)	1.46 (0.57, 3.78)	2.25 (1.16, 4.35) *
Antimony—Tertile 3 (0.36 μg/g)	1.12 (0.39, 3.25)	1.09 (0.58, 2.03)
Barium—Tertile 1 (1.26 μg/g)	REF	REF
Barium—Tertile 2 (3.09 μg/g)	1.15 (0.39, 3.39)	1.48 (0.82, 2.66)
Barium—Tertile 3 (15.49 μg/g)	2.33 (0.83, 6.55)	1.00 (0.54, 1.86)
Cadmium—Tertile 1 (0.09 μg/g)	REF	REF
Cadmium—Tertile 2 (0.16 μg/g)	1.04 (0.39, 2.80)	1.65 (0.90, 3.03)
Cadmium—Tertile 3 (0.62 μg/g)	1.60 (0.58, 4.35)	0.91 (0.49, 1.67)
Cesium—Tertile 1 (3.08 μg/g)	REF	REF
Cesium—Tertile 2 (4.13 μg/g)	0.39 (0.09,1.65)	1.17 (0.65, 2.10)
Cesium—Tertile 3 (8.75 μg/g)	2.98 (1.07, 8.25) *	0.55 (0.30, 1.01)
Cobalt—Tertile 1 (0.33 μg/g)	REF	REF
Cobalt—Tertile 2 (0.52 μg/g)	1.55 (0.55, 4.35)	0.79 (0.45, 1.40)
Cobalt—Tertile 3 (1.55 μg/g)	1.78 (0.59, 5.37)	1.25 (0.65, 2.39)
Lead—Tertile 1 (0.24 μg/g)	REF	REF
Lead—Tertile 2 (0.43 μg/g)	0.41 (0.10, 1.73)	1.09 (0.61, 1.95)
Lead—Tertile 3 (2.36 μg/g)	3.31 (1.09, 9.97) *	0.89 (0.46, 1.72)
Strontium—Tertile 1 (122.36 μg/g)	REF	REF
Strontium—Tertile 2 (201.59 μg/g)	0.47 (0.10, 2.16)	1.08 (0.60, 1.93)
Strontium—Tertile 3 (726.83 μg/g)	4.71 (1.34, 16.61) *	1.17 (0.59, 2.29)
Thallium—Tertile 1 (0.09 μg/g)	REF	REF
Thallium—Tertile 2 (0.13 μg/g)	1.34 (0.51, 3.53)	0.83 (0.46, 1.49)
Thallium—Tertile 3 (0.28 μg/g)	1.09 (0.39, 3.03)	1.25 (0.68, 2.27)
Tin—Tertile 1 (0.14 μg/g)	REF	REF
Tin—Tertile 2 (0.89 μg/g)	1.55 (0.59, 4.06)	0.64 (0.34, 1.21)
Tin—Tertile 3 (6.54 μg/g)	1.73 (0.41, 7.32)	0.82 (0.36, 1.90)
Tungsten—Tertile 1 (0.05 μg/g)	REF	REF
Tungsten—Tertile 2 (0.11 μg/g)	0.94 (0.27, 3.22)	0.92 (0.51, 1.65)
Tungsten—Tertile 3 (1.80 μg/g)	4.00 (1.11, 14.44) *	1.56 (0.82, 2.94)
Uranium—Tertile 1 (0.01 μg/g)	REF	REF
Uranium—Tertile 2 (0.02 μg/g)	0.98 (0.34, 2.89)	0.86 (0.46, 1.62)
Uranium—Tertile 3 (0.10 μg/g)	1.50 (0.58, 3.90)	0.77 (0.42, 1.39)

Note: All models were adjusted for age, seafood consumption within the past three days and cohort. (Nicaragua or NBCS). ***** Statistically significant at an alpha level of 0.05.

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
