# Peer review of "Urinary Metals Concentrations and Biomarkers of Autoimmunity among Navajo and Nicaraguan Men"

_ijerph, 2020, doi:10.3390/ijerph17155263_

Round 1
Reviewer 1 Report
Analysis of urinary concentrations of metals from Navajo and Nicaraguan men is presented, where associations with biomarkers of autoimmunity are examined. In general, the topic is interesting and the manuscript is well-written; yet, some issues have to be addressed in order to improve its clarity and applicability.
The contribution is not clear enough. For instance, why is it important to study Navajo and Nicaraguan men? Is there a real need to analyze these two groups (or it is just the availability of data)?
From the datasets (Navajo n=68, Nicaraguan n=47 and median age), it is evident that there are some disparities in data. How do these issues affect your results? For readers, it will be interesting to include more information for each individual (age, weight, years of work, etc.) since it affects directly the results. From a statistical viewpoint, the used datasets are good enough? reliable? Datasets from 2015 are still reliable/useful?
For clarity, please include flowcharts for the procedures carried out in sections 2.2-2.4.
Please add some graphs for the information presented in Tables 1-32 to make more evident trends, maxima and minima values, etc.
In the last paragraph of section 4, the limitations of your results are described, but the grade of impact of these issues is not discussed. For instance, it is said that “the small sample size and relatively low prevalence of positive outcomes (ANA and specific autoantibody reactivity) requires cautionary interpretation of statistical significance.”. How much could this fact affect your results? In simple words, how reliable are your results?
In the Conclusions section, the obtained results are somehow summarized, but they are important? Useful for what? They represent a problem? Can you indicate potential solutions to alleviate the problem? The applicability and importance of your findings are not clear enough.
Describe all the acronyms, e.g., NHANES
Improve quality of figures.
Author Response
Review 1:
Comment: Analysis of urinary concentrations of metals from Navajo and Nicaraguan men is presented, where associations with biomarkers of autoimmunity are examined. In general, the topic is interesting and the manuscript is well-written; yet, some issues have to be addressed in order to improve its clarity and applicability.
Response: Thank you very much.
Comment: The contribution is not clear enough. For instance, why is it important to study Navajo and Nicaraguan men? Is there a real need to analyze these two groups (or it is just the availability of data)?
Response: Both communities have health concerns that may be attributed to metals exposure and both are understudied. In both places, this is a first exploratory look at evidence of exposure among healthy individuals. Now we can conduct more targeted exposure assessments and examine associations with health. We did this for ANA as per autoimmune disease, a concern in the Navajo population in particular.
Comment: From the datasets (Navajo n=68, Nicaraguan n=47 and median age), it is evident that there are some disparities in data. How do these issues affect your results? For readers, it will be interesting to include more information for each individual (age, weight, years of work, etc.) since it affects directly the results. From a statistical viewpoint, the used datasets are good enough? reliable? Datasets from 2015 are still reliable/useful?
Response: Thank you for all of these questions and considerations. Yes, the metals concentrations are still a reliable source of information and will be important to publish. Five years is not too long a time to publish data on what we expect are chronic exposures and for which there are no active interventions. The relationship we observed between metals concentrations and ANA will not change. As for the information presented for each individual, we absolutely agree that individual age, weight, years of work, etc. are important considerations. The models include age as a predictor, which is likely associated with both exposure and outcome. However, unlike occupation, we did not have weight of the Navajo participants which could be related to both metals concentrations and outcome. We also worried about multiple comparisons on such a small, exploratory data set. We edited the Limitations section (line 383) to include the fact that we did not include weight as a variable in our models.
Comment: For clarity, please include flowcharts for the procedures carried out in sections 2.2-2.4.
Response: I sincerely apologize, I we did not make a flowchart of the methods section.
Comment: Please add some graphs for the information presented in Tables 1-32 to make more evident trends, maxima and minima values, etc.
Response: We do appreciate that this is a lot of information. We revamped Figure 1, creating two additional figures with error bars that show the geometric means and 95% confidence limits of the means for much of the data presented in Table 1, plus NHANES. We hope this is an improvement.
Comment: In the last paragraph of section 4, the limitations of your results are described, but the grade of impact of these issues is not discussed. For instance, it is said that “the small sample size and relatively low prevalence of positive outcomes (ANA and specific autoantibody reactivity) requires cautionary interpretation of statistical significance.”. How much could this fact affect your results? In simple words, how reliable are your results?
Response: Our results are reliable. We measured exposures and outcomes accurately and have high confidence in our findings. The statistical testing of significance is what we suggest readers interpret with caution. The small sample size and low outcome numbers are consistent with an exploratory study.
Comment: In the Conclusions section, the obtained results are somehow summarized, but they are important? Useful for what? They represent a problem? Can you indicate potential solutions to alleviate the problem? The applicability and importance of your findings are not clear enough.
Response: Thank you for these questions. We edited lines 402-403 to make explicit mention of the opportunity to reduce exposure to the metals for which very high concentrations were observed, as they may be associated with other adverse health outcomes that we did not study.
Comment: Describe all the acronyms, e.g., NHANES
Response: We went through the document to ensure that all acronyms are described. NHANES is described at first mention on line 180, and we added an explanation that did not previously exist on lines 135-136.
Comment: Improve quality of figures.
Response: We recreated Figure 1, resulting in two separate Figures that represent more information.
Reviewer 2 Report
This is an interesting, well-written paper on a relatively new topic in environmental health. It seems a bit random that these two different populations are studied in the same paper, but the authors do a good job of explaining their logic. Recommend publication after minor revisions:
Introduction
Lines 62-66: Here, it would be good to have a sentence or two broadly describing some of the hypothesized mechanisms linking metals and autoimmune response, even if not much has been published on this topic to date.
Methods
Are the urinary metals concentrations normally distributed or lognormally distributed? Perhaps the geometric means and geometric standard deviations should be reported in Table 1 instead of the arithmetic medians.
A mixed regression approach may have been appropriate, to account for intra-group (i.e., Navajo vs. Nicaragua) correlation. Was this attempted in the regression modeling, or in sensitivity analyses? It would be good to see how results compare. Can mixed regression results be shown in the main text or in supplementary information?
Somewhere in the statistical analysis methods section it should be stated explicitly whether or not weighted or unweighted NHANES 2011-2012 were used. Since the study men are fairly young, e.g., 23-51 years old, and healthy, it would be better to take a weighted subsample of NHANES men from this age group, without kidney disease, rather than all men >20 years old. NCHS provides helpful guidance on how to adjust the NHANES weights to analyze data from subsamples like this. Could these analyses be provided either in the main body or as supplemental analyses? This is important because the NHANES comparison is one focus of the paper, so we should be sure the comparison group is appropriate.
Using the NHANES weights would also allow for estimation of weighted geometric means and geometric standard deviations, which can be compared to the study geo. means and geo. SDs, for a more rigorous comparison than just comparing arithmetic medians with no measures of spread (beyond ranges). Also, it would be better to see geo. means (including weighted geo. means for the NHANES subsample) in Figure 1 with the geo. SDs as error bars, to give readers more information to make a rigorous comparison.
Since multiple metals are being considered from the same urine samples, it might be interesting to see the Pearson correlation coefficients between the metals (on the ln transformed urinary concentrations). This would give readers thinking about potential mixtures-type approaches information on whether or not colinearity is likely to be important with these data. This could be provided as a supplemental table.
Results
Is it possible that some of the resampled Nicaraguan sugarcane workers had kidney disease (e.g., CKD) that was not picked up in the workplace screening or was not reported on the questionnaire for some reason (i.e., fear of job loss, controversy surrounding Mesoamerican nephropathy? other?) How likely is such potential misclassification and how might it influence the study results?
Discussion
Consider reporting the urinary metals concentrations as ug/g creatinine instead of ug/g to avoid any confusion.
Table 1.
Can you specify “years” in the median age row?
Can the analytical limits of detection for each element be listed in a footnote? The % detected is not that interesting without seeing the LODs. Also, even though all the samples were analyzed in the same lab at NCEH, the NHANES LODs should be reported somewhere in the paper too, so readers have all the information they need to compare detection frequencies.
Figure 1.
Can you use ug/g creatinine instead of ug/g in the figure legend?
Author Response
Review 2:
Comment: This is an interesting, well-written paper on a relatively new topic in environmental health. It seems a bit random that these two different populations are studied in the same paper, but the authors do a good job of explaining their logic. Recommend publication after minor revisions:
Lines 62-66: Here, it would be good to have a sentence or two broadly describing some of the hypothesized mechanisms linking metals and autoimmune response, even if not much has been published on this topic to date.
Response: Thank you for the comments. We appreciate it is a bit random to bring these groups together and especially appreciate that you understand our logic. We added five lines of text (57-59, 64-66) and four references summarizing some possible mechanisms for the metal-autoimmune pathway. Thank you.
Comment: Are the urinary metals concentrations normally distributed or lognormally distributed? Perhaps the geometric means and geometric standard deviations should be reported in Table 1 instead of the arithmetic medians.
Response: Most of the metals were lognormally distributed. However, the median and geometric means are very similar. We calculated both. Since the logistic regression model evaluates exposure in percentiles, we decided that median values are more consistent with that approach, and that the median and range may be more easily interpreted by the readers (including the populations we study). However, triggered by your question and the comments of reviewers 1 and 3, we redid Figure 1 to represent the geometric mean values and confidence limits of our data, and compared those with the same data from NHANES which are now represented in two separate figures. Edits in lines 217 -224 explain these changes.
Comment: A mixed regression approach may have been appropriate, to account for intra-group (i.e., Navajo vs. Nicaragua) correlation. Was this attempted in the regression modeling, or in sensitivity analyses? It would be good to see how results compare. Can mixed regression results be shown in the main text or in supplementary information?
Response: Thank you for this suggestion. We did not attempt a linear mixed effects model. There were so many comparisons being made, and we did attempt to estimate correlations (see below), but with our small sample sizes in each group, we landed on the decision to include cohort in our models along with confounding variables.
Comment: Somewhere in the statistical analysis methods section it should be stated explicitly whether or not weighted or unweighted NHANES 2011-2012 were used. Since the study men are fairly young, e.g., 23-51 years old, and healthy, it would be better to take a weighted subsample of NHANES men from this age group, without kidney disease, rather than all men >20 years old. NCHS provides helpful guidance on how to adjust the NHANES weights to analyze data from subsamples like this. Could these analyses be provided either in the main body or as supplemental analyses? This is important because the NHANES comparison is one focus of the paper, so we should be sure the comparison group is appropriate.
Response: These comments are super helpful for consideration in our future analyses of metals in comparison to NHANES. As it were we used the NHANES values published in the 4th report and did not curate a dataset. We agree that would be a worthwhile analysis. Regarding your concern that our population data may be skewed to the younger ages compared with NHANES participants, and that because NHANES participants may have other diseases including kidney disease the populations are not exactly comparable, we cannot disagree. However, the consequence of poor kidney function on urinary metals concentrations is not certain. And, we think the likelihood is very low that NHANES participants of older age and/or with advanced kidney disease would significantly skew the median values of all males over 20 years of age.
Comment: Using the NHANES weights would also allow for estimation of weighted geometric means and geometric standard deviations, which can be compared to the study geo. means and geo. SDs, for a more rigorous comparison than just comparing arithmetic medians with no measures of spread (beyond ranges). Also, it would be better to see geo. means (including weighted geo. means for the NHANES subsample) in Figure 1 with the geo. SDs as error bars, to give readers more information to make a rigorous comparison.
Response: Thank you. We decided to compare median values of our data (with their ranges published) with the published median values in NHANES, and then present tertile concentrations in results. However, as mentioned in response to a comment above, we did calculate geometric means and created new figures including their confidence limits. We are convinced that future comparisons of our data with NHANES should include a data curation, with weighting by age and really appreciate this suggestion. Again, this was an exploratory study.
Comment: Since multiple metals are being considered from the same urine samples, it might be interesting to see the Pearson correlation coefficients between the metals (on the ln transformed urinary concentrations). This would give readers thinking about potential mixtures-type approaches information on whether or not colinearity is likely to be important with these data. This could be provided as a supplemental table.
Response: We agree that this is an intriguing question. We attempted to evaluate metals mixtures and clustering but the models were significantly underpowered. Given the shortcomings of our sample size, we indicate that additional research may be warranted on metals mixtures (lines 377-378) and include our small sample size as the reason for our inability to evaluate interactions between metals as a limitation (lines 380-381).
Comment: Is it possible that some of the resampled Nicaraguan sugarcane workers had kidney disease (e.g., CKD) that was not picked up in the workplace screening or was not reported on the questionnaire for some reason (i.e., fear of job loss, controversy surrounding Mesoamerican nephropathy? other?) How likely is such potential misclassification and how might it influence the study results?
Response: It is very unlikely that the Nicaraguan workers had kidney disease. We collected blood when we collected the urine samples, and estimated GFR to determine that their kidney function was within normal ranges.
Comment: Consider reporting the urinary metals concentrations as ug/g creatinine instead of ug/g to avoid any confusion.
Response: We went through the manuscript and included creatinine each time we reported units of ug/g to avoid any confusion by the reader.
Comment: Table 1. Can you specify “years” in the median age row?
Response: Yes, thank you. We included “years” in the median age row.
Comment: Table 1. Can the analytical limits of detection for each element be listed in a footnote? The % detected is not that interesting without seeing the LODs. Also, even though all the samples were analyzed in the same lab at NCEH, the NHANES LODs should be reported somewhere in the paper too, so readers have all the information they need to compare detection frequencies.
Response: In response to this comment we created a Supplementary table of the LODs provided by the CDC Lab and published with the NHANES report. We refer to the table in the text lines 148-149.
Comment: Figure 1. Can you use ug/g creatinine instead of ug/g in the figure legend?
Response: Yes, we edited the figure legend to include “creatinine.”
Reviewer 3 Report
Comments to Author
Ms. Ref. No.: ijerph-840805
Title: Urinary metals concentrations and biomarkers of autoimmunity among Navajo and Nicaraguan men
Among the possible environmental contributors to autoimmune disease, heavy metal pollution exposure has received very little attention. The ability of metals to induce or accelerate an autoimmune disease is dependent on speciation, route of exposure, dose, the genetic makeup, overall health, age, and gender of the exposed hosts.
The current study attempts to use data from two geographically distinct groups to characterize exposure to metals among healthy Nicaraguan and Navajo men of working age, evaluate biomarkers of autoimmunity (ANA and specific autoantibodies) in each group, and finally examine the relationship between metals exposure and biomarkers of autoimmunity in the pooled population. I found the paper to be overall well written and much of it to be well described.
But I have some specific comments are as follows.
- Figure 1, Standard error bar should be provided in each bar plot. Lack of clarity in Figure 1.
- Table 3, Provide the note of “REF” below the table.
- Line 400-406, The “XX” should be instead by one of the authors of this paper.
- I am also afraid of accurate effects due to small size samples and small selection of autoantibodies.
- Recommend author to read a relative reference “Metals and Autoimmune Disease” (DOI: 10.1007/978-3-642-27786-3_972-2).
Author Response
Review 3:
Comment: Among the possible environmental contributors to autoimmune disease, heavy metal pollution exposure has received very little attention. The ability of metals to induce or accelerate an autoimmune disease is dependent on speciation, route of exposure, dose, the genetic makeup, overall health, age, and gender of the exposed hosts.
The current study attempts to use data from two geographically distinct groups to characterize exposure to metals among healthy Nicaraguan and Navajo men of working age, evaluate biomarkers of autoimmunity (ANA and specific autoantibodies) in each group, and finally examine the relationship between metals exposure and biomarkers of autoimmunity in the pooled population. I found the paper to be overall well written and much of it to be well described. But I have some specific comments are as follows.
Response: Thank you.
Comment: Figure 1, Standard error bar should be provided in each bar plot. Lack of clarity in Figure 1.
Response: Thank you. We revised Figure 1 to include two new figures that provide a error bar.
Comment: Table 3, Provide the note of “REF” below the table.
Response: We edited the Table 3 Title to more clearly explain REF, rather than making a footnote.
Comment: Line 400-406, The “XX” should be instead by one of the authors of this paper.
Response: Yes, we think that actual names were removed by the editors to protect author identity.
Comment: I am also afraid of accurate effects due to small size samples and small selection of autoantibodies.
Response: Yes, we agree and refer to the study as “hypothesis generating” and “exploratory” several times in the manuscript.
Comment: Recommend author to read a relative reference “Metals and Autoimmune Disease” (DOI: 10.1007/978-3-642-27786-3_972-2).
Response: Thank you. We used this reference in response to a comment by reviewer 2, and have edited lines 57-59 of text with reference to this important book chapter.
Round 2
Reviewer 1 Report
All my comments and concerns have been properly addressed. The Reviewer recommends manuscript acceptance.